# The Ability of the Hopkins Symptom Checklist-5 to Identify Generalized Anxiety Disorder and Major Depressive Disorder in the General Population

**DOI:** 10.3390/ijerph22050698

**Published:** 2025-04-28

**Authors:** Benedicte Kirkøen, Ragnhild Elise Ørstavik, Anne Reneflot, Jens Christoffer Skogen, Børge Sivertsen, Ann Kristin Skrindo Knudsen

**Affiliations:** 1Department of Mental Health, Norwegian Institute of Public Health, 0473 Oslo, Norway; ragnhild.orstavik@tidsskriftet.no (R.E.Ø.); anne.reneflot@fhi.no (A.R.); 2Department of Health Promotion, Norwegian Institute of Public Health, 5015 Bergen, Norway; jens.christoffer.skogen@fhi.no (J.C.S.); borge.sivertsen@fhi.no (B.S.); 3Centre for Evaluation of Public Health Measures, Norwegian Institute of Public Health, 0473 Oslo, Norway; 4Alcohol and Drug Research Western Norway, Stavanger University Hospital, 4068 Stavanger, Norway; 5Department of Research & Innovation, Helse-Fonna HF, 5525 Haugesund, Norway; 6Department of Disease Burden, Norwegian Institute of Public Health, 5015 Bergen, Norway; ann.kristin.knudsen@fhi.no

**Keywords:** Hopkins Symptom Checklist, Composite International Diagnostic Interview, anxiety, depression, validation

## Abstract

**Background**: The Hopkins Symptom Checklist (HSCL) is a widely used measure of anxiety and depression symptoms. The short form HSCL-5 is especially suitable for large population-based studies, but its ability to detect mental disorders in the general population remains unknown. The aim of the study was to assess how well the HSCL-5 identified cases of generalized anxiety disorder (GAD) and major depressive disorder (MDD) measured by the Composite International Diagnostic Interview (CIDI) 5.0 and to find the optimal sex-specific cut-off levels of the HSCL-5. **Methods**: Participants from the population-based Trøndelag Health Study (HUNT) in Norway were recruited for the current study. Between April and September 2020, 1343 participants (64% women) aged 20–65 years completed the CIDI, followed by the HSCL-5. The overall agreement between the HSCL-5 and GAD or MDD measured by CIDI was examined with Receiver Operator Characteristics (ROC) analysis. Sensitivity, specificity, positive predictive value (PPV), and negative predictive values (NPV) for different cut-off levels were assessed. **Results**: The area under the curve for GAD or MDD was 0.90 (CI 95% = 0.85–0.95) for women and 0.85 (CI 95% = 0.68–1.00) for men. For women, a cut-off level of ≥1.80 had the best balance between sensitivity (85%) and specificity (84%), while the corresponding numbers were ≥2.00, 73%, and 93% for men. The global PPV was 21%, while the NPV was 99%. **Conclusions**: The HSCL-5 has high sensitivity and specificity for identifying cases of GAD or MDD. In the current study, the positive predictive value of HSCL-5 was low.

## 1. Introduction

Mental disorders are a leading cause of disability worldwide [1,2]. Recently, an increase in the prevalence of symptoms of anxiety and depression has been found in many countries, especially among young adults [3,4,5,6,7]. This observation has sparked a debate as to whether an increase in self-reported symptoms also reflects an increase in the true prevalence of mental disorders.

For the planning of health promotion and work to prevent mental disorders, accurate prevalence estimates of mental disorders are crucial. Structured diagnostic interviews conducted in population-based surveys represent the gold standard for assessing the prevalence of mental disorders, but such assessments are costly and time-consuming. Hence, there is a global lack of such studies [8], while shorter questionnaires assessing symptoms of mental disorders have become increasingly important in monitoring the mental health of the population [9]. To approximate prevalence estimates of disease burden, cut-off values have been developed to identify possible cases of mental disorders from questionnaires. A valid and reliable assessment of the case-finding abilities of these questionnaires is of the utmost importance in epidemiological research settings. In Norway, large population-based health surveys have been conducted on a regular basis for several decades. A common questionnaire used in these surveys is the Hopkins Symptom Checklist (HSCL) [10].

Since its development in the 1950s, the HSCL [10,11] has become established in clinical research as a measure of symptoms of mental disorders. The first versions of 40 and 90 items were later reduced to HSCL-25, which captures the two most frequent mental disorders: anxiety (measured by 10 items) and depression (15 items). Whether HSCL-25 can discriminate between the two is, however, still under debate [12,13]. The case-finding abilities of HSCL-25 have been tested against structured diagnostic interviews in several studies since the 1990s, indicating that HSCL-25 has good abilities to detect depression but varying results for the detection of anxiety [12,14,15].

In epidemiological surveys, long questionnaires may cause lower response rates, which compromise the generalizability of results [16,17]. Thus, shorter but still valid questionnaires are in demand. In a Norwegian sample, Tambs and Moum identified five questions from HSCL-25 and showed that this short form had good psychometric properties [18]. While studies show that the HSCL-5 has demonstrated good validity when compared to the total HSCL-25 [19,20], to our knowledge, only one study has compared HSCL-5 to a structured diagnostic interview [21]. The authors showed that HSCL-5 is a valid tool for detecting depression in primary care. To our knowledge, no study has validated HSCL-5 against a structured diagnostic interview in the general population or evaluated the ability of HSCL-5 to detect anxiety measured by a structured diagnostic interview. Therefore, the ability of HSCL-5 to detect anxiety and depression in the general population remains unknown. This knowledge would be an important contribution to the literature, as several large population-based studies use HSCL-5 (e.g., [22,23,24,25,26,27]). Moreover, various studies compare HSCL to different subgroups of anxiety and depressive disorders. A Norwegian study found that HSCL-25 correlated most strongly with depression, panic disorder, and generalized anxiety disorder (GAD) [12]. While phobias are more prevalent among anxiety disorders, the study showed that HSCL-25 did not predict phobias well. Major depressive disorder (MDD) is the most prevalent among depression disorders [28,29], and when HSCL-25 is reduced to the short form HSCL-5, items relating to panic disorder are excluded. Consequently, the current study aims to test the ability of HSCL-5 to detect GAD and MDD.

Further, different studies recommend different cut-off levels for HSCL-5 (an average score on the HSCL-5 of ≥1.80 or ≥2.00). Additionally, men and women report different levels of anxiety and depression symptoms (e.g., [3]), which may indicate differences in their experience and reporting of symptoms. Consequently, a recent Spanish study suggests that optimal cut-of levels of HSCL should be stratified by sex [14], a finding further supported by a study of Norwegian students [30]. This highlights the need for new knowledge on appropriate cut-off levels for HSCL-5 in both sexes.

The aim of the current study was to test the ability of HSCL-5 to identify generalized anxiety disorder (GAD) and major depressive disorder (MDD) through the Composite International Diagnostic Interview (CIDI) [31,32] in a population-based sample and determine the sensitivity, specificity, positive predictive value, negative predictive value. Further, the study aimed to identify optimal cut-off levels of the HSCL-5 for women and for men.

## 2. Materials and Methods

**Sample**. Participants in the Trøndelag Health Study (HUNT) were invited to participate in a structured diagnostic interview survey. HUNT is a longitudinal, population-based health study in Nord-Trøndelag and Trøndelag (last wave only), two counties in mid-Norway, inviting every county resident above 20 years to participate (for more information, see [33]). Invitees to the current study were sampled among HUNT participants aged 20 to 65 years from Trondheim city. HUNT in Trondheim had a participation rate of 43% [33]. The diagnostic interview survey is registered at ClinicalTrials.gov (identifier: NCT04661228) and approved by the Regional Committee for Medical Research Ethics (2017/28/REK-midt). Participants invited to the diagnostic interview between April to September 2020 (*n* = 4300) were also asked to complete the HSCL-5, and 1379 (32%) completed both the interview and HSCL-5. All participants gave written informed consent.

## 3. Procedure and Measures

**Procedure.** The invitees received postal letters with information about the project, followed by an SMS that informed them how to sign up for the diagnostic interview. One SMS was sent to remind those not responding, and among those who registered for the study, up to four attempts were made to schedule the interview (three by phone and one final by SMS). All participants received a GBP 30 (NOK 300) gift card. The interviews were conducted face-to-face (*n* = 278) or by telephone (*n* = 1101) by trained and certified interviewers. Participants completed the diagnostic interview, followed by the questionnaire.

**CIDI.** Participants completed the Composite International Diagnostic Interview 5th version (CIDI) [31,32], a standardized interview developed by the World Health Organization (WHO). CIDI 5.0 assesses the 30-day, 12-month, and lifetime prevalence of several mental and substance use disorders according to the criteria in the Diagnostic and Statistical Manual of Mental Disorders 5th edition (DSM-5) [34] and the International Classification of Diseases 10th edition (ICD-10) [35]. It has demonstrated high validity and reliability for mental disorder assessment in the general population [31,36]. The mental disorders were operationalized according to diagnostic algorithms developed by the World Mental Health Survey (WMHS) for use on CIDI 5.0. In the current study, we aimed to compare CIDI-derived diagnoses with the HSCL-5. The HSCL-5 has a 2-week timeframe, and in order to approximate a comparable timeframe, only the prevalence of diagnoses based on the last 30 days from CIDI was utilized.

Case definitions. The 30-day mental disorder was defined as the presence of a mental disorder during the 30 days prior to the interview (yes/no). As HSCL-5 was designed to measure symptoms of anxiety and depression, the following mental disorders were included: generalized anxiety disorder (GAD) and major depressive disorder (MDD) as individual diagnoses, and GAD or MDD combined, any anxiety or depressive disorder (including generalized anxiety disorder, panic disorder, specific phobia, agoraphobia, social anxiety disorder, major depressive disorder, bipolar type I and II disorders).

**HSCL-5.** The Hopkins Symptom Checklist (HSCL-5) was used to assess symptoms of anxiety and depression. The HSCL-5 consists of 5 items scored on a Likert scale from 1 (‘not bothered’) to 4 (‘very bothered’). Participants are asked to rate to which degree they are bothered by the different symptoms for the past two weeks. An average score on the HSCL-5 of ≥1.80 or ≥2.00 is commonly used as a cut-off for identifying possible cases of mental disorders and corresponds well with the cut-off of ≥1.75 on the 25-item version (HSCL-25) [19,21].

Sex, age, level of education, and whether the respondent was living with a partner were reported by participants in the questionnaire. Participants’ level of education was categorized as (a) high school level, (b) lower degree of higher education (≤four years at a university or college), and (c) higher degree of higher education—(≥four years at a university or college). Living with a partner was coded “yes” if the participant reported to be married or living with someone in a marriage-like relationship and “no” otherwise.

## 4. Statistical Analyses

**Factor analyses.** We conducted an explorative factor analysis and an orthogonal varimax rotation to determine whether the HSCL-5 was best understood as one or more factors in our sample.

**Criterion validity.** The overall agreement between the HSCL-5 and the criterion standard CIDI generalized anxiety disorder (GAD) or major depressive disorder (MDD), or GAD and MDD separately, were examined with Receiver Operator Characteristics (ROC) analysis. The ROC curve represents sensitivity and specificity for all different cut-off levels and was used to determine the optimal cut-off level overall for women and men separately and for participants below and above 30 years. Sensitivity is the ability of HSCL-5 to detect a true positive case, and specificity is an expression of how well the test detects a true negative case [37]. In combination, sensitivity and specificity indicate the diagnostic accuracy of HSCL-5 compared to CIDI and are therefore important markers of criterion validity. The area under the ROC curve (AUC) is an index of the goodness of the scale. The AUC was estimated with 95% CI. Positive and negative predictive values for different cut-off levels were also assessed. All analyses were conducted in STATA, version 17.0.

## 5. Results

**Descriptives.** In total, 1379 participants completed both the CIDI interview and HSCL-5. Participants included and excluded from the analyses are shown in Figure 1. Four participants had missed only one item in HSCL-5. For these, the mean score was calculated based on the four valid items. Three participants had missing information on bipolar type II disorder. These three participants were categorized as having “no affective disorder”, as they also had no MDD or bipolar type I disorder. Twenty participants had missing diagnostic information on any anxiety disorder and were categorized as “no”, as they had no positive diagnosis on the remaining other anxiety disorders. The final sample consisted of 1343 participants (31% of the invitees). The demographics of the sample and the number of people with different CIDI diagnoses are shown in Table 1.

**Factor analyses.** Prior to performing factor analysis, the suitability of data for factor analysis was assessed. The Kaiser–Meyer–Olkin value was 0.86, and Bartlett’s test of sphericity was significant, *p* < 0.01, both indicating the suitability of data for factor analysis. The explorative factor analyses showed the two first eigenvalues of 2.85 and 0.05, yielding a ratio between them of 57, indicating essential unidimensionality. Only the first factor had an eigenvalue above 1, meaning that according to Kaiser’s criterion (eigenvalues > 1), HSCL-5 is best understood as one factor. Consequently, HSCL-5 is treated as one factor in all further analyses. Table 2 shows the factor loadings for all items on one factor. The internal consistency of the whole HSCL-5 scale was high (α = 0.87; with all items contributing to increase the estimated scale reliability).

**Criterion validity.** The mean HSCL-5 score was 1.32 (SD = 0.47) for the whole sample, 2.32 (SD = 0.68) among those with GAD, and 2.28 (SD = 0.84) among those with MDD. Measured by HSCL-5, 222 individuals (16.5%) scored above ≥1.80 (19% of the women and 12% of the men). Of the 222 with a score above ≥1.80, 46 (21%) had a GAD or MDD diagnosis. Out of those with a GAD or MDD diagnosis in CIDI, 46 (82%) had an HSCL-5 score above ≥1.80 (83% of those with a GAD diagnosis, 77% of those with MDD diagnosis).

The ROC curves for GAD or MDD are presented separately for men and women in Figure 2. The point-estimate for AUC was marginally better for women (AUC= 0.90, CI 95% = 0.85–0.95) than for men (AUC = 0.85, CI 95% = 0.68–1.00). The separate ROC curves for GAD and MDD are presented in Figure 3 and Figure 4.

The sensitivity, specificity, positive predictive value (PPV), and negative predictive value (NPV) of different cut-off levels of HSCL-5 are shown in Table 3, for GAD or MDD and for GAD and MDD separately (for all cut-off levels, see Appendix A; for other diagnoses, see Appendix A). In the current sample, no one scored at 1.75, and that cut-off level is therefore not reported. Overall, HSCL had the best balance between sensitivity (82%) and specificity (86%) for GAD or MDD using ≥1.80 as the cut-off point. For women, a cut-off level of ≥1.80 had the best balance between sensitivity (85%) and specificity (84%). For men, a cut-off level of ≥2.00 had the best balance between sensitivity (73%) and specificity (93%) (see Appendix A). For participants ≤30 years old (*n* = 434), a cut-off level of ≥2.00 had the best balance between sensitivity (83%) and specificity (88%) (see Appendix A). For participants >30 (*n* = 909), a cut-off level of ≥1.60 had the best balance between sensitivity (85%) and specificity (80%).

Sensitivity and specificity for GAD and MDD separately are presented in Table 3. The sensitivity for any affective or anxiety disorder was 0.45, while the specificity was 0.88 (for more details, see Appendix A).

## 6. Discussion

The importance of the validity and reliability of measures of mental health are receiving renewed interest as recent studies show an increase in the prevalence of young people reporting symptoms of anxiety and depression [3,4,5,6,7]. To the best of our knowledge, the current study is the first to address to what degree HSCL-5 captures common mental disorders (generalized anxiety disorder (GAD) or major depressive disorder (MDD)) diagnosed with a structured interview (CIDI) in the general population. Our results indicate that HSCL-5 discriminates well between people with and without GAD or MDD, with HSCL-5 showing high sensitivity and specificity, supporting the criterion validity of the scale. The negative predictive value was also found to be high. However, the positive predictive value of HSCL-5 in the general population was low.

Our results confirm that HSCL-5 may be best understood as a unidimensional measure [12,13]. This may not necessarily reflect the quality of the questionnaire per se but rather reflect that symptoms of anxiety and depression often co-occur and may be, to some extent, overlapping [38,39]. The area under the ROC curve above 0.80 indicates that HSCL-5 discriminates well between people with and without GAD or MDD and between people with or without GAD. For MDD, the area under the ROC curve was 0.86 for women but 0.75 for men. While the latter is lower, all levels above 0.75 are considered to be of clinical utility [40].

In the study by Rodríguez-Barragán and colleagues (2023), the sensitivity and specificity for depression were 78% and 73% for HSCL-5 [21], similar to the sensitivity found in the current study for the detection of MDD (77%). However, we found a higher specificity (85%). Previous studies comparing the 25-item version to structured diagnostic interviews report a sensitivity between 70% and 88% and a specificity of between 77% and 85% for depression or mood disorders, and a sensitivity of 43–50% and a specificity of 83% for anxiety disorders [12,14,15]. We show a similar sensitivity and specificity for depression and specificity for anxiety compared to previous studies of the full-scale HSCL-25. However, our results show a higher sensitivity for HSCL when compared to GAD, while the sensitivity when HSCL was compared to any anxiety disorder was low. This might indicate that HSCL-5 discriminates well between people with and without generalized anxiety disorder (GAD) but is less effective in detecting people with other anxiety disorders. Importantly, the different studies are not directly comparable as they include different diagnoses of anxiety and depression.

For the total study population in the current study, the optimal cut-off level was ≥1.80, which provides a good balance between sensitivity (0.82) and specificity (0.86) for GAD or MDD. This means that HSCL-5 correctly identifies 82% of the people diagnosed with GAD or MDD according to CIDI and correctly identifies 86% of the people with no diagnoses. Importantly, we identify different optimal cut-off levels for women and for men. For men, the cut-off level of ≥2 provided the best balance between sensitivity (0.73) and specificity (0.93). The cut-off for HSCL-5 was first recommended to be ≥2, based on a study comparing HSCL-5 to the full HSCL scale [19]. However, a recent study comparing HSCL-5 to a structured diagnostic interview in a clinical setting found that the cut-off of ≥1.80 fitted well for the total patient population and ≥2 for men for the detection of depression [21]. These sex-specific cut-off values are confirmed in the current study, for the prediction of both GAD and MDD in the general population. This provides confidence in our results. One should note, however, that in the current study, no participants had a score of 1.75, which is the recommended cut-off level for HSCL-25. Consequently, we were unable to investigate whether 1.75 or 1.80 received the best balance. We observed that a higher cut-off level had the best balance for participants below 30 years old, and a lower level were most suitable for participants above 30 years. This is interesting to note as many studies report an increase in the prevalence of symptoms of anxiety and depression among young adults. Future studies of young adults should consider to take this into account.

Our choice of optimal cut-off level is based on the use of HSCL-5 for epidemiological purposes, and therefore, the cut-off with the best balance between sensitivity and specificity was chosen [41]. However, which cut-off level is considered optimal depends on the purpose for its use. If HSCL-5 is used with the aim of identifying all people with a mental disorder, one may choose a lower cut-off level with higher sensitivity. On the other hand, if HSCL-5 were to be used for screening in the general population, with limited resources available for follow-up diagnostic interviews, a cut-off level with higher specificity (at the cost of lower sensitivity) may be preferable.

The ability of a questionnaire to detect a disorder depends both on the characteristics of the questionnaire and on the prevalence of the disease in the population of interest. While the sensitivity, specificity, and negative predictive value of HSCL-5 were all high in our study, the positive predictive value (PPV) was low. This means that when a person scores above the cut-off on HSCL-5, the probability of having an actual diagnosis of GAD or MDD, according to CIDI, is low (21%). Our results indicate that if 100 people in a population are tested with HSCL-5, 16 of them will score above the cut-off of ≥1.80, but only 3 out of the 16 (21%) will have a diagnosis of GAD or MDD. To put it differently, the remaining 13 persons who score above the cut-off are unlikely to be diagnosed with GAD or MDD (even though their symptoms might be burdening). The current study offers an important perspective on the interpretation of the observed increases in people scoring above the cut-off for anxiety and depression using shorter questionnaires.

Importantly, PPV (and NPV) is highly connected to the actual prevalence of mental disorders in the population. Using a test with the same sensitivity and specificity, the PPV will increase with increasing prevalence in the population tested. This is illustrated in Table 4, which shows the PPV and the NPV in the current sample as well as estimates of two hypothetical samples with higher prevalence of GAD and MDD, of 8.4% and 16%, respectively. As can be seen in Table 4, in a population with a prevalence of 8.4% or 16% of GAD or MDD, PPV tells us that 36% and 53% of people who score above the cut-off are likely to have a diagnosis of GAD or MDD. Thus, in the latter example, the test result is more correct, even though the characteristics of the test are the same. Consequently, HSCL-5 will have higher positive predictive value in populations with a higher prevalence of mental disorders (e.g., primary care settings) compared to the general population.

## 7. Strengths and Limitations

To the best of our knowledge, this is the first study to address to what degree HSCL-5 captures common mental disorders (GAD or MDD) diagnosed with a structured interview (CIDI) in the general population. Our study has several strengths: First, the data were collected as part of an internationally recognized population-based health study (HUNT). Second, while previous studies examining the corresponding abilities of HSCL have only referred participants with a score above the cut-off level to the diagnostic interview—the current study invited all participants to complete both CIDI and the HSCL. Thus, the results can be used to determine to what degree this short questionnaire can be applied to determine the true prevalence of mental disorders in a population. Third, both examinations were completed on the same day, ensuring partly overlapping time frames for symptom assessments.

The results must be interpreted in light of six limitations: First, although this study was population-based, there was substantial attrition (participation rate is 31%). While attrition in follow-up studies is a substantial problem when examining exposure–outcome associations and prevalence rates, it might be less problematic for cross-sectional association studies such as those applied in the current project [42]. The prevalence of GAD (2.7%) and MDD (2.3%) was lower than what has been reported in previous cross-sectional studies from Norway [43], but this might be due to the limited time frame (30 days) we applied in the current study. Second, while the HSCL-5 asks participants to report symptoms for the past 2 weeks, CIDI asks participants to report symptoms from the past month. Thus, the two measures do not have the same timeframe, which may influence the results. Third, the CIDI interviews were conducted by phone and face-to face. Previous studies have shown, however, that this is unlikely to have any effects of the results [44]. Fourth, data were collected during the COVID-19 pandemic. This might have had some implications, although we believe that any effects would be small as most of the data collections were conducted by phone, and this part of Norway suffered smaller consequences with regard to disease severity, lockdowns, etc. Fifth, interviews and questionnaires were administered on the same day and always in this order (interviews first). Thus, subjects might be influenced by their answers in the interview when filling out the HSCL questionnaire. Sixth, these results based on a Norwegian sample may not be fully generalizable to other regions and cultural settings.

## 8. Conclusions

The current study is the first to evaluate the case-finding abilities of Hopkins Symptom Checklist (HSCL)-5 against a diagnostic interview in a general population. The results show that HSCL-5 has high sensitivity and specificity and thus identifies people with diagnoses of GAD or MDD well in the general population. The optimal cut-off level for HSCL-5 in the current study was ≥1.80 (and ≥1.80 for women and ≥2.00 for men). The ability of a questionnaire to detect a disorder depends on the characteristics of the test as well as the prevalence of the disease in the population of interest. In the general population, the positive predictive value of HSCL-5 was low.

## Figures and Tables

**Figure 1 ijerph-22-00698-f001:**
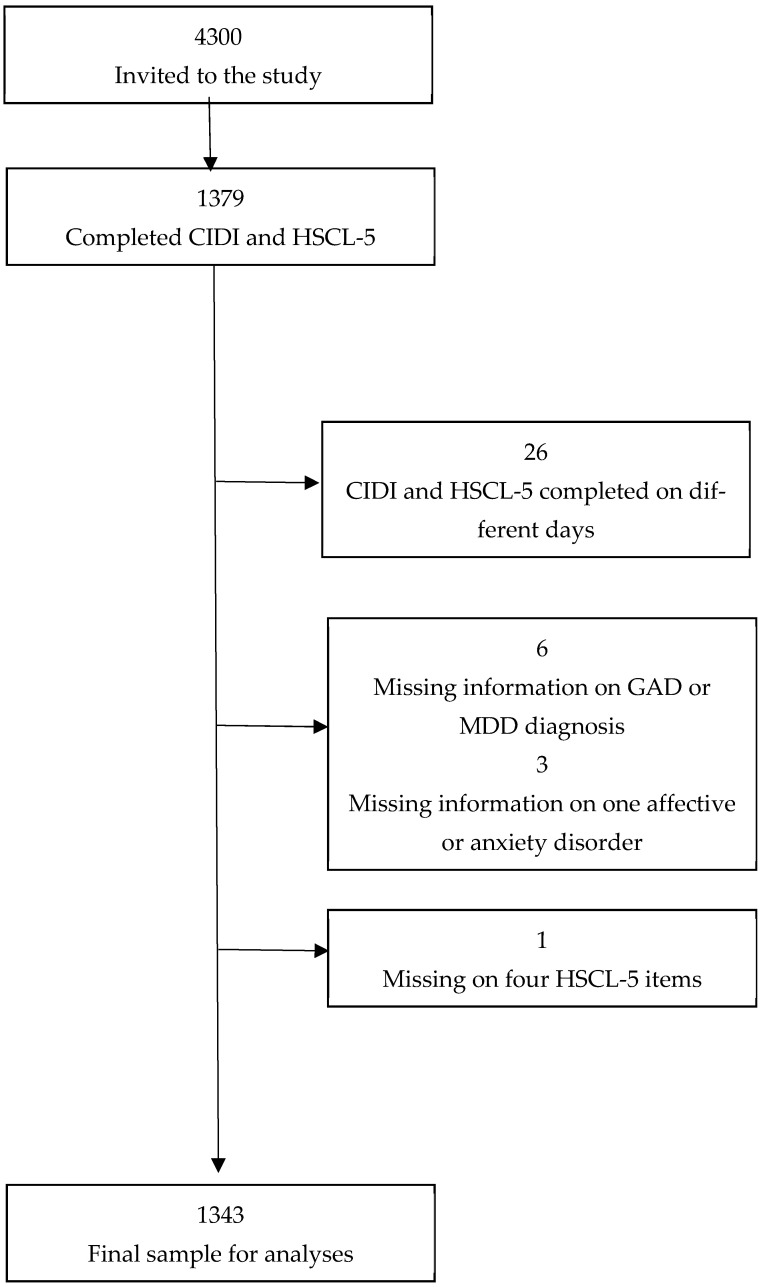
Flowchart of participants in the study.

**Figure 2 ijerph-22-00698-f002:**
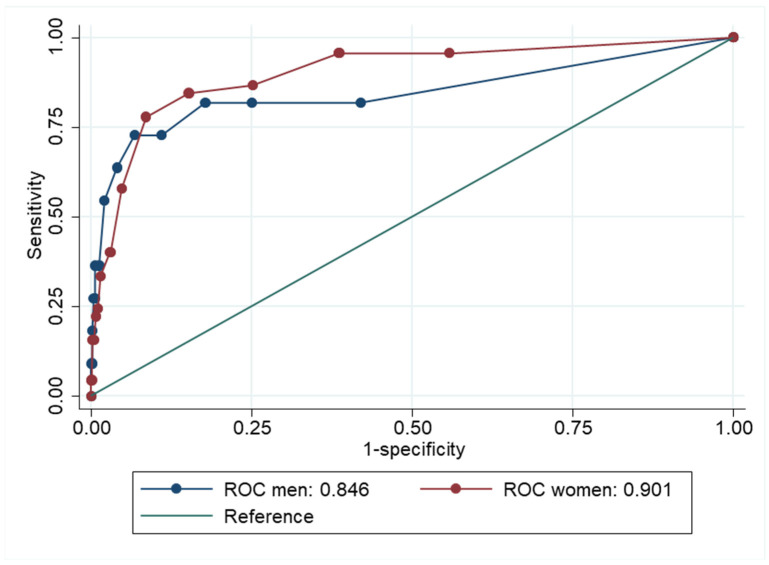
Receiver Operator Characteristics (ROC) curve showing how well Hopkins Symptom Checklist-5 compares to generalized anxiety disorder (GAD) or major depressive disorder (MDD) for men and women.

**Figure 3 ijerph-22-00698-f003:**
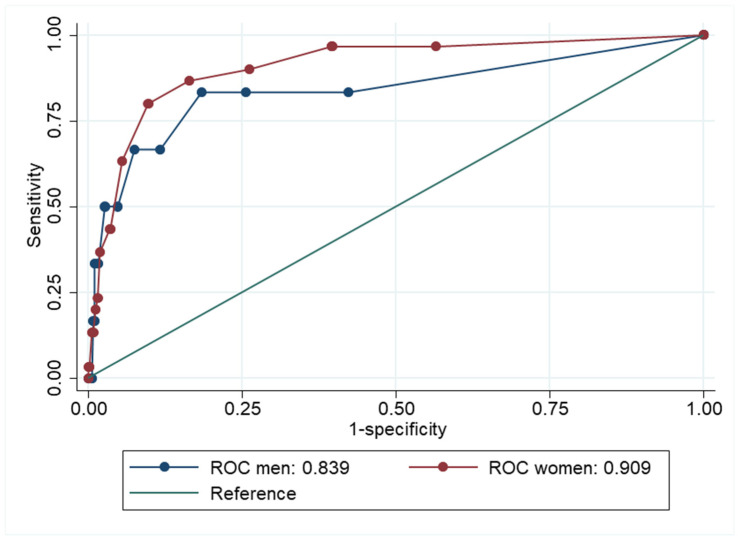
Receiver Operator Characteristics (ROC) curve showing how well Hopkins Symptom Checklist-5 compares to generalized anxiety disorder (GAD) for men and women.

**Figure 4 ijerph-22-00698-f004:**
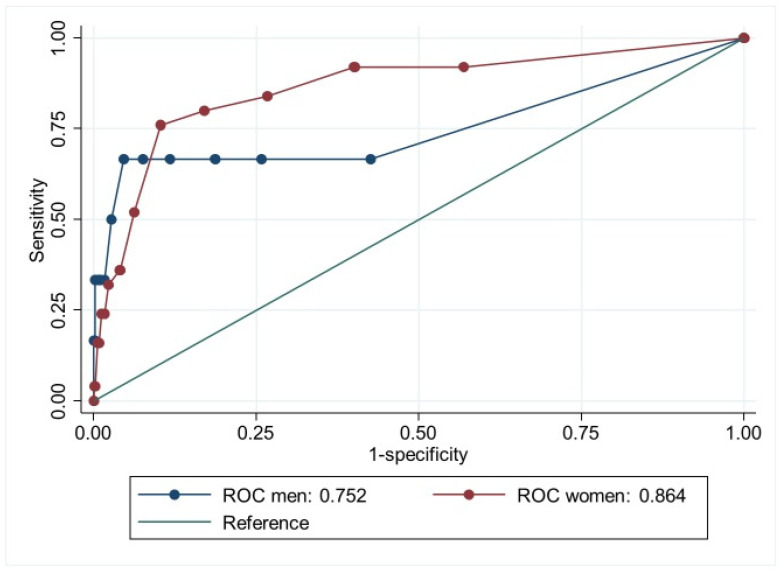
Receiver Operator Characteristics (ROC) curve showing how well Hopkins Symptom Checklist-5 compares to major depressive disorder (MDD) for men and women.

**Table 1 ijerph-22-00698-t001:** Descriptives of the sample.

Demographic Information	
Women, *n* = (%)	860 (64%)
Age, mean	39 (SD = 12, range 20–66)
Higher education, (lower or higher degree of higher education)	1036 (77%)
Diagnoses From CIDI, 30 Days	
GAD or MDD *, *n* = (%)	56 (4.2%)
Generalized anxiety disorder	36 (2.7%)
Major depressive disorder	32 (2.3%)
Any affective or anxiety disorder **	172 (13%)
Any anxiety disorder	154 (11.5%)
Any affective disorder	36 (2.7%)

CIDI = Composite International Diagnostic Interview. * Generalized anxiety disorder (GAD) or major depressive disorder (MDD). ** Including generalized anxiety disorder, panic disorder, specific phobia, agoraphobia, social anxiety disorder, major depressive disorder, bipolar type I and II disorders.

**Table 2 ijerph-22-00698-t002:** Factor loadings of the 5 items in Hopkins Symptom Checklist-5.

Item	Factor Loading
Feeling fearful	0.83
Nervousness or shakiness inside	0.83
Feeling hopeless about the future	0.77
Feeling blue	0.78
Worry too much about things	0.85

**Table 3 ijerph-22-00698-t003:** Sensitivity and specificity of Hopkins Symptom Checklist-5 (HSCL) at different cut-off levels for different CIDI diagnoses (30 days).

		CIDI Diagnoses		
		GAD or MDD *	GAD	MDD
HSCL cut-off ≥ 1.80	Sensitivity	0.82	0.83	0.77
	Specificity	0.86	0.85	0.85
	PPV	0.21	0.13	0.10
	NPV	0.99	0.99	0.99
HSCL cut-off ≥ 2.00	Sensitivity	0.77	0.78	0.74
	Specificity	0.92	0.91	0.90
	PPV	0.29	0.19	0.16
	NPV	0.98	0.99	0.99
HSCL cut-off ≥ 2.25	Sensitivity	0.43	0.44	0.38
	Specificity	0.97	0.97	0.96
	PPV	0.41	0.27	0.20
	NPV	0.97	0.98	0.98

CIDI = Composite International Diagnostic Interview. * Generalized anxiety disorder (GAD) or major depressive disorder (MDD), PPV = positive predictive value, NPV = negative predictive value.

**Table 4 ijerph-22-00698-t004:** Positive predictive value and negative predictive value in the current sample and two hypothetical populations with different prevalences of generalized anxiety disorder (GAD) or major depressive disorder (MDD).

	Current Sample	Hypothetical Population I	Hypothetical Population II
Prevalence of GAD or MDD, % (*n*=)	4.2% (56)	8.4% (113)	16% (215)
Sensitivity of HSCL-5 cut-off ≥ 1.80	0.82	0.82	0.82
Specificity of HSCL-5 cut-off ≥ 1.80	0.86	0.86	0.86
Positive predictive value (PPV)	0.21	0.36	0.53
Negative predictive value (NPV)	0.99	0.98	0.96

## Data Availability

Norwegian data protection regulations and GDPR impose restrictions on the sharing of individual participant data. However, researchers may gain access to survey participant data by contacting the publication committee (anne.reneflot@fhi.no). Approval from the Norwegian Regional Committee for Medical and Health Research Ethics (https://helseforskning.etikkom.no, accessed on 10 February 2025) is a pre-requirement for access to the data. The dataset is administrated by the HUNT databank, and guidelines for access to data are found at https://www.ntnu.edu/hunt/data (Accessed on 10 February 2025). The study protocol and the informed consent form are available on the two homepages of the project: https://www.ntnu.no/hunt/forskningsprosjekt (Accessed on 10 February 2025) and https://www.fhi.no/cristin-prosjekter/aktiv/diagnosebasert-undersokelse-psykiske-lidelser-og-ruslidelser/ (Accessed on 10 February 2025).

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
