# Peer review of "The Ability of the Hopkins Symptom Checklist-5 to Identify Generalized Anxiety Disorder and Major Depressive Disorder in the General Population"

_ijerph, 2025, doi:10.3390/ijerph22050698_

Round 1
Reviewer 1 Report
Comments and Suggestions for Authors
- How have you chosen Statistical analyses?You have applied explorative factor analysis,Receiver Operator Characteristics (ROC) analysis. Are they enough? is HSCL-5 reliable and valid? is HSCL-5 used as a scale somewhere? -Is CIDI 5.0 used widely in Norway?is it a valid and reliable tool for Norwegian people? Why have you chosen CIDI 5.0? but not another tool? -Where can we download CIDI 5.0 items? How many does CIDI 5.0 have ? you may give the link to reach them in the study or give the items of CIDI 5.0 in the study. -Line 120: which items does HSCL-5 consists?any link to see them? -Explain why do you apply sensitivity and specificity?what do they define?Ranges of them?....
Reviewer 2 Report
Comments and Suggestions for Authors
Applied research methodology must be improved:
1. Present tested assumptions for FA (which rotation is applied, which FA is applied - PCA, PAf, ML... If Oblique rotation is applied pattern matrix has to be presented), KMO, Bartlet´s test of sphericity has to be shown
2. for DV descriptive statistics have to be presented, not only SD and MEan, but: present skew, kurtosis, std errors, and KS- because FA is a parametric test
3. testing a difference between subsamples using a ROC to determine a statistical predictive impact can not be a primary statistical approach, but only as an additional approach. Regression analysis (linear or not linear or better Neural Networks- multilayer perceptron) has to be applied with a level of the stat. Significance p<0,01 due to having big samples - increased type one error)
Differences between values based on sensitivity and specificity, i.e., PPV and NPV, are not strong parameters for determining generalizations.
Round 2
Reviewer 1 Report
Comments and Suggestions for Authors
-I could not open https://www.hcp.med.harvard.edu/wmhcidi/.
Please , give link to reach these questions.
It give mesagges below.
Thank you for visiting the WMH-CIDI web page.
The World Mental Health Surveys team no longer supports the DSM-IV CIDI.
You can access the DSM-IV CIDI instrument at the National Comorbidity Survey website: https://www.hcp.med.harvard.edu/ncs/replication.php
CIDI-5 is now available, but restricted to the WMH Survey initiative members.
Author Response
Thank you very much for reviewing our manuscript.
I am sorry that the link could not be opened. But as the message states, CIDI-5 is restricted to the World Mental Health (WMH) Survey initiative members. CIDI-5 is managed and owned by World Mental Health Survey and is subject to their access guidelines. Access to the full CIDI-5 questionnaires is restricted to ensure proper administration and interpretation.
Therefore, we regret to inform you that we are unable to provide a full list of the items in the manuscript or provide a link to them.
Consequently, important studies that have used CIDI, (E.g. Kessler et al., 2005) provides no link to access the questions. See for instance the paper listed below
Kessler, R. C.; Chiu, W. T.; Demler, O.; Walters, E. E., Prevalence, severity, and comorbidity of 12-month DSM-IV disorders in the National Comorbidity Survey Replication. Archives of general psychiatry 2005, 62, (6), 617-627.